# Gene-Based Association Tests Using New Polygenic Risk Scores and Incorporating Gene Expression Data

**DOI:** 10.3390/genes13071120

**Published:** 2022-06-22

**Authors:** Shijia Yan, Qiuying Sha, Shuanglin Zhang

**Affiliations:** Department of Mathematical Sciences, Michigan Technological University, 1400 Townsend Drive, Houghton, MI 49931, USA; shijiay@mtu.edu (S.Y.); qsha@mtu.edu (Q.S.)

**Keywords:** PRS, TWAS, gene-base association studies

## Abstract

Recently, gene-based association studies have shown that integrating genome-wide association studies (GWAS) with expression quantitative trait locus (eQTL) data can boost statistical power and that the genetic liability of traits can be captured by polygenic risk scores (PRSs). In this paper, we propose a new gene-based statistical method that leverages gene-expression measurements and new PRSs to identify genes that are associated with phenotypes of interest. We used a generalized linear model to associate phenotypes with gene expression and PRSs and used a score-test statistic to test the association between phenotypes and genes. Our simulation studies show that the newly developed method has correct type I error rates and can boost statistical power compared with other methods that use either gene expression or PRS in association tests. A real data analysis figure based on UK Biobank data for asthma shows that the proposed method is applicable to GWAS.

## 1. Introduction

To date, conventional genome-wide association studies (GWAS) have been successfully applied to identifying the association of genetic variants with phenotypes. However, despite its many successes, there are two major challenges for GWAS: one is missing the heritability of complex diseases due to polygenic effects [1,2,3]; the other is the ambiguous biological interpretation of its findings, because some identified genetic variants are not in protein-coding regions.

Many alternative methods have been developed to handle these challenges. The International Schizophrenia Consortium (ISC) proposed a polygenic risk score (PRS) [4], which is now widely used in assessing the genetic liability to phenotypes [5]. Studies show that PRS not only can be applied to predict disease [6], but can also be used in gene-based association tests [7]. Moreover, there has been increased interest in integrating expression quantitative trait loci (eQTL) studies and GWAS to improve complex trait mapping. PrediXcan [8] and transcriptome-wide association studies (TWAS) [9] are two of the most widely used integrative methods for testing the associations between phenotypes and gene-expression values predicted from SNP genotyping or sequencing data. PrediXcan and TWAS offer increased power over traditional GWAS methods and facilitate the biological interpretation of their discoveries.

The polygenic risk score (PRS) is a sum of the trait-associated alleles across many genetic loci, typically weighted by effect sizes estimated from a GWAS. Although PRS-type methods can provide higher statistical power in gene-based association studies, they may suffer from great uncertainty in PRS estimation, with imperfect choices of effect-size estimates [10]. PrediXcan [8] and TWAS [9] integrate GWASs with eQTL data to discover candidate genes that are associated with phenotypes. Both PrediXcan [8] and TWAS [9] use a weighted burden test, and the weights are the cis-effects of the SNPs on the gene expressions derived from eQTL datasets [11]. Therefore, these methods are not suitable in situations in which SNPs influence phenotypes directly and are not associated with gene expression [9]. Studies show that TWAS retains high power when the expression mediates between SNPs and phenotypes, but has very-low-to-moderate power when SNPs directly and independently affect gene expression and phenotypes [12].

Taking the advantage of the methods involving the use of PRS and the methods involving the integration of GWAS with eQTL data in gene-based association studies, we develop a powerful gene-based association method leveraging both gene expression measurements and PRS. We also propose two new weights for PRS. The aim of the proposed methods is to improve upon the standard PRS method and the TWAS-type method in gene-based association tests. In our study, we use a generalized linear model to associate a phenotype with gene expression and PRS. Through simulation studies, we evaluate both the type I error rates and the powers of the proposed methods and compare the power of the new methods with other methods that use either gene expression data or PRS in gene-based association tests under different scenarios. Our simulation studies show that the proposed methods have correct type I error rates and are either the most powerful methods, or at least comparable with the most powerful methods.

## 2. Methods

In our gene-based association study, we assumed that individual-level phenotypes and genotypes were available. Suppose there are n individuals; each individual has a phenotype and genotypes of M SNPs in a gene. For the ith individual, let $y_{i}$ and $x_{i}={\left(x_{i1},\ldots ,x_{iM}\right)}^{T}$ denote the phenotype and genotypes in the gene, where i=1,…,n. Then, $X={\left(x_{1}^{},\ldots ,x_{n}^{}\right)}^{T}$ is the genotype matrix. In the following sections, we first give a brief review of the TWAS method [9]; next, we introduce the standard PRS and our new PRSs; finally, we describe a powerful gene-based association method leveraging both gene-expression measurements and PRSs.

### 2.1. TWAS

TWAS estimates gene expression based on an additional eQTL dataset with $n_{e}$ unrelated individuals. Let $ge_{i}$ denote the expression level of the gene. Assume that the gene expression is a linear model of the following genotype scores: $ge_{i}=\sum_{m=1}^{M}W_{m}x_{im}+\varepsilon_{i}$ for $i=1,2,\ldots ,n_{e}$, where $W_{m}$ is the cis-effect of SNP m on gene expression and $\varepsilon_{i}$ is the noise. Based on the linear model, elastic net [13] is used to obtain the estimate of $W_{m}$. Next, on a test set with n unrelated individuals, the gene expression of the ith individual can be predicted by the M SNP genotype of the ith individual $x_{i}={\left(x_{i1},\ldots ,x_{iM}\right)}^{T}$, that is, $E_{i}=\sum_{m=1}^{M}W_{m}x_{im}=W^{T}x_{i}$, where $W={\left(W_{1},\ldots ,W_{M}\right)}^{T}$ and i=1,…,n.

For a trait of interest, TWAS applies a generalized linear regression model to test for association between the trait and predicted expression by using one of the asymptotically equivalent Wald, score, and likelihood ratio tests [11]. In this paper, we use the score test [14] for TWAS and use pre-calculated weights to construct the predicted gene expression corresponding to a given tissue. The pre-calculated weights are available at Gusev_Lab [8,9] (http://gusevlab.org/projects/fusion/; accessed on 2 January 2022).

### 2.2. Newly Developed LD-Adjusted PRSs

The standard PRS of the ith individual in a gene is given by $PRS_{i}=\sum_{m=1}^{M}{\hat{\beta }}_{m}x_{im}={\hat{\beta }}^{T}x_{i}$, where ${\hat{\beta }}_{m}$ is the estimated genetic effect of the mth SNP on the phenotype and can be obtained from the summary statistics of a GWAS [15]. In fact, PRS can be viewed as a weighted sum of genotypes in a gene $PRS_{i}=\sum_{m=1}^{M}w_{m}x_{im}=w^{T}x_{i}$, where $w={\left(w_{1},\ldots ,w_{M}\right)}^{T}$. In the standard PRS, the weight $w_{m}$ is given by the estimated effect size ${\hat{\beta }}_{m}$ for the mth SNP. Good choices of $w_{m}$ should satisfy two conditions: (1) $\left|w_{m}\right|$ should be large if the mth SNP is strongly associated with the phenotype, and (2) $w_{m}$ can reflect the directions of the association. Based on these two conditions, we develop two new PRSs. Let $T_{m}$ be the score test statistic to test whether the mth SNP is associated with a phenotype. We can define new PRSs based on the following two weights: (1) $w_{m}=T_{m}$, the score test statistic for the mth SNP, and (2) $w_{m}=\mathrm{sign}\left(T_{m}\right)T_{m}^{2}$, the squared score-test statistic with its sign. Note that the score-test statistic $T_{m}$ can be obtained from the Z-score based on the GWAS summary statistics. If Z-score is not available, but the *p*-value is available in GWAS summary statistics, we can obtain the absolute value of the score test statistic T by $\left|T\right|=\Phi^{-1}\left(1-p/2\right)$, where Φ is the standard normal cumulative distribution function; the sign of T is the same as the sign of the corresponding ${\hat{\beta }}_{m}$. Corresponding to the three kinds of weight, we have three PRSs: (1) PRS_B_ with $w_{m}={\hat{\beta }}_{m}$, (2) PRS_T_ with $w_{m}=T_{m}$, and (3) PRS_Q_ with $w_{m}=\mathrm{sign}\left(T_{m}\right)T_{m}^{2}$.

For constructing PRSs, Baker et al. [10] proposed a LD-adjusted PRS. The presence of markers in LD gives a larger contribution to the PRS than a single or uncorrelated marker [10]. Instead of using LD pruning [16] to remove the LD for the standard PRS, we account for LD by using the LD-adjusted PRS with some modifications.

Let R denote the sample correlation matrix of genotypes in a gene. Baker et al. used ${\tilde{x}}_{i}=R^{-1/2}x_{i}$ to replace $x_{i}$ in PRS to adjust for LD between SNPs. If we let $e_{1},\ldots ,e_{M}$ and $\lambda_{1}\geq \cdots \geq \lambda_{M}$ denote the eigenvectors and corresponding eigenvalues of R, the eigenvectors $e_{1},\ldots ,e_{L}$ represent new orthogonal axes corresponding to decreasing variability of the genotype data. We can then write $R^{-1/2}$ as $R^{-1/2}=\sum_{l=1}^{M}e_{l}e_{l}^{T}/\sqrt{\lambda_{l}}$. Since very small values of $\lambda_{l}$ can make $R^{-1/2}$ unstable, we propose to use the following approach to calculate $R^{-1/2}$. Let L denote the smallest number such that $\sum_{l=1}^{L}\lambda_{l}/\sum_{l=1}^{M}\lambda_{l}\geq 0.999$, then we only use the first L components to calculate $R^{-1/2}$, that is, $R^{-1/2}\approx \sum_{l=1}^{L_{1}}e_{l}e_{l}^{T}/\sqrt{\lambda_{l}}$. After we calculate $R^{-1/2}$, based on Baker et al.’s approach [10], we use the adjusted genotypes ${\tilde{x}}_{i}=R^{-1/2}x_{i}$ to calculate PRS, that is, $PRS_{i}=w^{T}{\tilde{x}}_{i}$. We adjust all three PRSs using the method mentioned above in the following studies.

### 2.3. Association Test Leveraging Both Gene Expression Measurements and PRSs

We assumed that we had GWAS summary statistics for a phenotype and an additional eQTL data set or pre-calculated weights for gene expression, such as the weights provided at Gusev_Lab [8,9] (http://gusevlab.org/projects/fusion/; accessed on 2 January 2022). Our proposed method is based on the following model: $y_{i}=\beta_{0}+\beta_{1}E_{i}+\beta_{2}PRS_{i}+\varepsilon_{i}$ if the phenotype is quantitative, and $\mathrm{logit}\left(P\left(y_{i}=1|E_{i},PRS_{i}\right)\right)=\beta_{0}+\beta_{1}E_{i}+\beta_{2}PRS_{i}$ if the phenotype is qualitative for i=1,…,n. To test whether a gene is associated with a phenotype, the null hypothesis is given by $H_{0}:\beta_{1}=\beta_{2}=0$. We use a score test with a chi-squared distribution $\chi_{2}^{2}$ to test the null hypothesis.

We denote our methods by TWAS-PRSs. Corresponding to the three kinds of weights in the PRSs, there are three TWAS-PRSs: (1) TWAS-PRS_B_ with $w_{m}={\hat{\beta }}_{m}$, (2) TWAS-PRS_T_ with $w_{m}=T_{m}$, and (3) TWAS-PRS_Q_ with $w_{m}=\mathrm{sign}\left(T_{m}\right)T_{m}^{2}$.

## 3. Comparison of Methods

We compared the performance of TWAS-PRSs with the other four methods: TWAS [9] and three PRS-based methods, PRS_B_, PRS_T_, and PRS_Q_. The three PRS-based methods are based on the model $y_{i}=\beta_{0}+\beta_{1}PRS_{i}+\varepsilon_{i}$ if the phenotype is quantitative, or $\log\mathrm{it}\left(P\left(y_{i}=1|PRS_{i}\right)\right)=\beta_{0}+\beta_{1}PRS_{i}$ if the phenotype is qualitative. To test whether a gene is associated with the phenotype, the null hypothesis is $H_{0}:\beta_{1}=0$. The score-test statistic with $\chi_{1}^{2}$ distribution is used for the association test. Corresponding to the three PRSs, we have three PRS-based association tests: PRS_B_, PRS_T_, and PRS_Q_. If there are covariates, we adjust the phenotypes for the covariates by a linear regression and use the residuals as new phenotypes in the corresponding association tests [17,18].

## 4. Simulations

The COPD gene dataset [19] was used in the simulation studies. This dataset contains genotypes of 5430 unrelated individuals on 630,860 SNPs. We chose three genes: GTF2H2 (gene1), ZNF514 (gene2), and RP11-426C22 (gene3), which contain 15, 37, and 64 SNPs, respectively. We use the program fastPHASE [20] to infer haplotype phases for the 5430 individuals to obtain 10,860 haplotypes. To generate the genotype of an individual, we randomly chose two haplotypes from 10,860 haplotypes. We obtained weights $W={\left(W_{1},\ldots ,W_{M}\right)}^{T}$ for gene expression from the TWAS website (http://gusevlab.org/projects/fusion/; accessed on 2 January 2022).

To generate gene expression, we used the model $E_{i}=\sum_{m=1}^{M}W_{m}x_{im}+e_{i}$, where $e_{i}\sim N(0,\sigma^{2})$, $\sigma^{2}=W^{T}\mathrm{cov}(X)W$, and $W={\left(W_{1},\ldots ,W_{M}\right)}^{T}$. To generate the phenotype of an individual, we used a model similar to that described by Liang et al. [21]:(1)$$y_{i}=\beta (aE_{i}+\sum_{j=1}^{c}x_{ij})+\varepsilon_{i},$$
where $E_{i}$ is the gene expression for the ith individual, $x_{i1},\ldots ,x_{ic}$ are genotypes of c causal variants that are directly associated with the phenotype, a is a constant weight to indicate how the phenotype is influenced by gene expression compared with those directly associated causal variants, β is the total effect of gene expression and directly associated causal variants, and $\varepsilon_{i}\sim N\left(0,1\right)$.

To generate a qualitative trait, we used a liability threshold model based on a continuous phenotype (quantitative trait). An individual was defined as affected if the individual’s phenotype was at least one standard deviation larger than the phenotypic mean. This yielded a prevalence of 16% for the simulated disease in the general population. In this study, we performed 1000 simulations with a significance level of 0.05.

We generated individual-level genotype and phenotype data for n=5000 unrelated individuals. To obtain GWAS summary statistics (${\hat{\beta }}_{m}$ and $T_{m}$), we additionally generated genotypes and phenotypes with sample size N=5000, 10,000, and 20,000, respectively. We considered a proportion of causal variants in each gene, prop=0.2, and 0.3, then the total number of causal variants c was the ceiling of M·porp, c=ceiling(M·porp). We used a=1 in the simulation study.

We also considered using different gene expression weights to generate $E_{i}$. Let $m_{\max}=\arg\max\{W_{1},\ldots ,W_{M}\}$ and $W_{\max}={\left(0,\ldots ,0,W_{m_{\max}},0,\ldots ,0\right)}^{T}$. Let $W^{*}={\left(W_{1}^{*},\ldots ,W_{M}^{*}\right)}^{T}=W-W_{\max}$, $m_{\max}^{*}=\arg\max\{W_{1}^{*},\ldots ,W_{M}^{*}\}$, $W_{\max}^{*}={\left(0,\ldots ,0,W_{m_{\max}^{*}}^{*},0,\ldots ,0\right)}^{T}$, and $W_{h}=W_{\max}+W_{\max}^{*}$. The two weights, $W_{\max}$ and $W_{h}$, were used in our simulations.

Based on Equation (1), the two weights ($W_{\max}$ and $W_{h}$), and the three genes (gene1, gene2, and gene3), we considered a total of six models for every particular setting: Models 1 to 3 with $W=W_{\max}$ for gene1, gene2, and gene3; and Models 4 to 6 with $W=W_{h}$ for gene1, gene2, and gene3.

## 5. Simulation Results

### 5.1. Type I Error Rates

To evaluate the type I error rates of the seven methods, we considered different sample sizes of GWAS data sets (5000, 10,000, and 20,000) and different genes (gene1, gene2, and gene3). We first generated the phenotypes and genotypes under the null hypothesis; next, we calculated the GWAS summary statistics based on the GWAS data sets; finally, we calculated the Type I error rates for the seven methods. For the 1000-replicates samples, the 95% confidence interval (CI) for the estimated type I error rates of 5%wais (0.0365,0.0635). Table 1 summarizes the estimated type I error rates of the seven methods under different scenarios. We can see that all of the estimated type I error rates were within the corresponding 95% CIs for the different sample sizes of the GWAS data sets and different genes, which indicates that all of the seven tests were valid.

### 5.2. Powers

We compared the powers of the seven tests with different values of the total effect size β, different sample size for the GWAS N, and different proportions of causal variants prop for quantitative traits. Figure 1, Figure 2 and Figure 3 show the power comparisons for the sample sizes N=5000, 10,000, 20,000 with prop=0.2. Appendix A also show the power comparisons for the sample sizes N=5000, 10,000, 20,000 with prop=0.3. These figures show similar power patterns. In general, TWAS-PRSs are more powerful than PRSs, and PRSs are more powerful than TWAS; among the three different PRSs, PRS_Q_ performs better than PRS_B_ and PRS_T_; PRS_B_ and PRS_T_ perform similarly. When the sample size for the GWAS N increases, the power of PRSs and TWAS-PRSs increase. The powers also increase as a proportion of increase in the causal variants. We also evaluated the powers of the seven tests for qualitative traits with different models and settings. Similar results can be found in Appendix A for the qualitative traits. In conclusion, TWAS-PRSs leveraging the information from eQTL and GWAS showed a better performance. In the following section, we apply the seven methods to the UK Biobank data.

## 6. Application to UK Biobank Data

### 6.1. UK Biobank Data

The UK Biobank [22] is a population-based cohort study with a wide variety of genetic and phenotypic information [23]. We applied the seven methods to analyze the UK Biobank [22] dataset for asthma. In this study, we only considered SNPs located in autosomal chromosomes and subjects with white British ancestry. The quality control of the samples and variants was performed by plink2. We filtered out the variants with minor allele frequency (MAF) of less than 0.05 and with *p*-values of the Hardy–Weinberg equilibrium (HWE) exact test below $10^{-6}$. We exclude variants with missing call rates exceeding 0.01 and dosage certainty of less than 0.9. We deleted samples with missingness exceeding 0.01.

The asthma cases were defined based on field code 6152_8 (doctor-diagnosed asthma), the International Classification of Diseases version-10 (ICD10) J45 (asthma)/J46 (severe asthma), and self-reported asthma [24]. Field 6152 is a summary of the distinct main diagnosis codes the participants recorded across all their hospital visits. The non-asthmatic controls were defined as individuals free from field code 6152_8 and field code 6152_9 (doctor-diagnosed allergic diseases), ICD10 J45/J46/J30 (hay fever)/L20 (dermatitis and eczema), and self-reported asthma/hay fever/eczema/allergy/allergy to house dust mites (HDMs). This strategy resulted in a broad definition of asthma, with 45,016 cases and 240,511 controls in the UK Biobank after quality control.

Since many thyroid diseases can lead to pulmonary problems [25,26], we considered using weights for gene expression based on the thyroid of GTEx v7. The pre-computed weights are available at: http://bogdan.bioinformatics.ucla.edu/software/twas/ (accessed on 2 January 2022). We used the weights estimated by BLUP, and only considered variants with both genotypes and gene-expression weights available. For each gene, we considered SNPs located between the gene boundary and ±500 kb.

### 6.2. Results

After pre-processing, there were 285,527 individuals and 9807 genes for the analysis. We considered age, sex, the first ten principal components, and the genotype array as the covariates in this study. We then adjusted the phenotype by the covariates using a linear regression model [17,18]. To compare the performances of the three TWAS-PRSs, we divided the 285,527 individuals into two sets with different sample sizes, corresponding to three settings: (1) N=2n; (2) N=n; and (3) 2N=n, where N is the sample size of the dataset to calculate the GWAS summary statistics and n is the sample size of the individual-level genotype and the phenotype dataset for the association test, and N+n=285,527. Since there were a total of 9807 genes on the 22 chromosomes, at 5% significance level, the Bonferroni threshold of $5.098\times 10^{-6}$ was used to determine the significant genes.

We applied the seven methods, TWAS, PRS_B_, PRS_T_, PRS_Q_, TWAS-PRS_B_, TWAS-PRS_T_, and TWAS-PRS_Q_, to the data set under different settings. Table 2 summarizes the number of genes identified by each method. Both PRS_Q_ and TWAS-PRS_Q_ identified more genes than the corresponding methods; PRS_T_ and TWAS-PRS_T_ identified almost the same number of genes as PRS_B_ and TWAS-PRS_B_, respectively; and TWAS identified the lowest number of genes. As the sample size of the individual-level dataset became lager, more genes were identified by all the methods. We also compared the number of identified genes that were reported in TWAS hub (http://twas-hub.org/; accessed on 2 January 2022), represented by the numbers in the parentheses in Table 2. It can be seen that PRS_Q_ and TWAS-PRS_Q_ identify more genes near the loci reported in TWAS hub than the corresponding methods. Overall, our proposed methods, PRS_Q_, and TWAS- PRS_Q_, are applicable to GWAS and perform better than TWAS and the corresponding methods.

## 7. Discussion

Gene expression is an important mechanism, since the regulatory variants influence complex traits through transcriptional regulation [27]. On the other hand, PRS is exploited to assess shared etiologies between phenotypes [15], which is a powerful tool in predictions and tests. In this research, we provide new weights for constructing PRS, which can boost the statistical power of using PRS in gene-based association tests. Furthermore, we propose the TWAS-PRS method, which can take both PRS and gene expression into consideration.

However, there are several limitations to the current study. Although the incorporation of gene-expression measurements will facilitate biological interpretation, we still cannot claim causality, for which experimental validations are required. Furthermore, since we did not consider trans-eQTLs, but only cis-eQTLs, many genes were not included in our study. When calculating gene expression, the choice of weights also influences the performance of our methods. Although we performed real data analysis using thyroid tissue for our asthma study, further studies are needed to assess which tissue could be more relevant to the pathogenesis of asthma, such as nasal or lung tissues [28].

In conclusion, we provided two additional weights to construct PRSs and compare their performances. We leveraged both gene-expression measurements and PRSs to fit a linear model and used a score test to test the associations between genes and phenotypes. The simulation studies showed that our proposed methods, PRS_Q_ and TWAS_Q_, can not only control type I error rates but also have higher power than the corresponding methods. Furthermore, the application of our proposed methods to the UK biobank data analysis shows that the proposed methods are applicable to real data GWAS and perform better than the corresponding methods.

## Figures and Tables

**Figure 1 genes-13-01120-f001:**
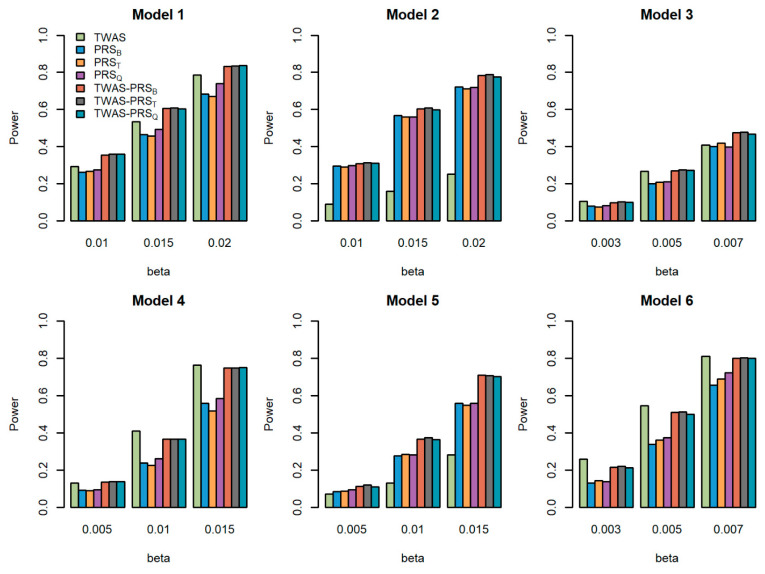
Powers of the seven tests versus the total effect size β for quantitative traits with N=5000. The proportion of causal variants is 0.2. Models 1–3 correspond to genes 1–3, for which we only used the eQTL with the largest weight to generate gene expression; Models 4–6 correspond to genes 1–3, for which we used the two eQTLs with the first two largest weights to generate gene expression.

**Figure 2 genes-13-01120-f002:**
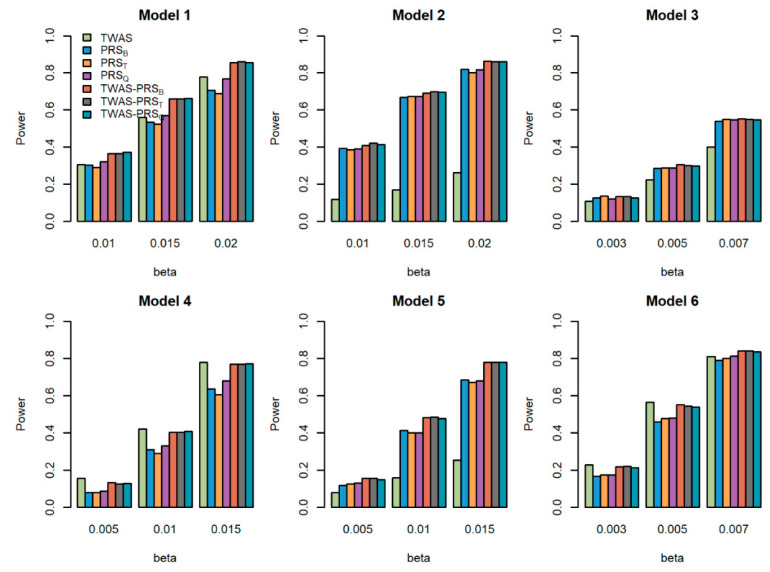
Powers of the seven tests versus the total effect size β for quantitative traits with N=10,000. The proportion of causal variants is 0.2. Models 1–3 correspond to genes 1–3, for which we only used the eQTL with the largest weight to generate the gene expression; Models 4–6 correspond to genes 1–3, for which we used two eQTLs with the first two largest weights to generate gene expression.

**Figure 3 genes-13-01120-f003:**
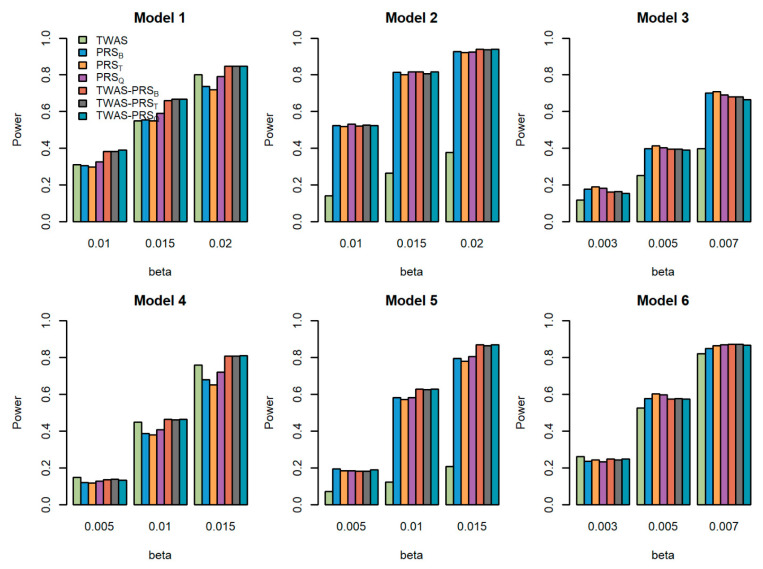
Powers of the seven tests versus the total effect size β for quantitative traits with N=20,000. The proportion of causal variants is 0.2. Models 1–3 correspond to genes 1–3, for which we only use the eQTL with the largest weight to generate gene expression; Models 4–6 correspond to genes 1–3, for which we use two eQTLs with the first two largest weights to generate gene expression.

**Table 1 genes-13-01120-t001:** Estimated type I error rates of the seven methods for different sample sizes of GWAS data sets (5000, 10,000, and 20,000) and different genes (gene1, geme2, and gene3). Type I error rates are evaluated using 1000-replicates sample at significance level of 0.05.

N	Gene	TWAS	PRS_B_	PRS_T_	PRS_Q_	TWAS-PRS_B_	TWAS-PRS_T_	TWAS-PRS_Q_
5000	1	0.044	0.056	0.062	0.057	0.056	0.057	0.058
	2	0.048	0.051	0.048	0.050	0.063	0.061	0.063
	3	0.046	0.042	0.045	0.045	0.049	0.051	0.050
10,000	1	0.044	0.055	0.057	0.051	0.060	0.063	0.058
	2	0.054	0.046	0.047	0.049	0.052	0.047	0.047
	3	0.050	0.052	0.054	0.056	0.060	0.057	0.046
20,000	1	0.043	0.049	0.047	0.047	0.054	0.051	0.055
	2	0.039	0.040	0.040	0.041	0.043	0.044	0.047
	3	0.040	0.042	0.039	0.047	0.040	0.042	0.042

**Table 2 genes-13-01120-t002:** The number of genes identified by seven methods under different settings. The numbers in the parentheses indicate the number of identified genes that are reported in TWAS hub (http://twas-hub.org/; accessed on 2 January 2022).

Setting	TWAS	PRS_B_	PRS_T_	PRS_Q_	TWAS-PRS_B_	TWAS-PRS_T_	TWAS-PRS_Q_
n=(1/2)N	47 (28)	190 (98)	198 (98)	218 (124)	198 (100)	195 (99)	212 (113)
n=N	65 (34)	257 (149)	249 (148)	258 (152)	249 (145)	247 (145)	268 (157)
n=2N	82 (43)	319 (185)	312 (186)	337 (203)	304 (186)	297 (185)	324 (205)

## Data Availability

Not applicable.

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
