# Peer review of "Gene-Based Association Tests Using New Polygenic Risk Scores and Incorporating Gene Expression Data"

_genes, 2022, doi:10.3390/genes13071120_

Round 1
Reviewer 1 Report
This manuscript is well written. The proposed method shows superior performance in simulation studies.
My only comment is to explain the abbreviation "TWAS".
Reviewer 2 Report
In this paper the authors develop a method: TWAS-PRS which combines TWAS analysis with PRS in order to identify genes related to a trait.
The main problem of the manuscript is that table 2 is missing from the material I downloaded. It is not included in the pdf or as an external table. Without table 2 I can't really assess the final results of the paper.
It is not clear to me if the PRS is defined for the specific region of the gene, or it is a genome-wide PRS. If it is specific, was some region around the gene chosen? Otherwise, it shouldn't be informative of the effect of the gene and just have the same value in all genes.
Figures 1-6 can be moved to supplementary material, maybe leaving figure 1-3 (or one of the three) for information but leaving the rest since the results are quite similar in all of them and there is not new information learned.
The chose of tissue for the asthma analysis it seems odd to me. Other papers looking at causal genes for asthma used gene expression data for lung and blood (https://www.nature.com/articles/s42003-021-02227-6). A comparison with the results of this paper, showing which genes are common and which are different would be interesting and informative to whether the analysis identifies the correct genes.
It seems that the authors choose 1-2 snps to run the TWAS analysis from the weights in FUSION. Why do not use all that have weight (and overlap) for the gene? The amount of SNPs used for TWAS analysis can increase TWAS performance considerably
